

# Data augmentation for Arabic text classification: a review of current methods, challenges and prospective directions

Samia F. Abdhood[1,2], Nazlia Omar[1] and Sabrina Tiun[1]

[1] Center for Artificial Intelligence Technology, Faculty of Information Science and Technology, Universiti Kebangsaan Malaysia, Bangi, Selangor, Malaysia
[2] Faculty of Computers and Information Technology, Hadhramout University, Almukalla, Hadhramout, Yemen

Corresponding author
Samia F. Abdhood,
abdhood1987@gmail.com

## ABSTRACT

The effectiveness of data augmentation techniques, *i.e.*, methods for artificially creating new data, has been demonstrated in many domains, from images to textual data. Data augmentation methods were established to manage different issues regarding the scarcity of training datasets or the class imbalance to enhance the performance of classifiers. This review article investigates data augmentation techniques for Arabic texts, specifically in the text classification field. A thorough review was conducted to give a concise and comprehensive understanding of these approaches in the context of Arabic classification. The focus of this article is on Arabic studies published from 2019 to 2024 about data augmentation in Arabic text classification. Inclusion and exclusion criteria were applied to ensure a comprehensive vision of these techniques in Arabic natural language processing (ANLP). It was found that data augmentation research for Arabic text classification dominates sentiment analysis and propaganda detection, with initial studies emerging in 2019; very few studies have investigated other domains like sarcasm detection or text categorization. We also observed the lack of benchmark datasets for performing the tasks. Most studies have focused on short texts, such as Twitter data or reviews, while research on long texts still needs to be explored. Additionally, various data augmentation methods still need to be examined for long texts to determine if techniques effective for short texts are also applicable to longer texts. A rigorous investigation and comparison of the most effective strategies is required due to the unique characteristics of the Arabic language. By doing so, we can better understand the processes involved in Arabic text classification and hence be able to select the most suitable data augmentation methods for specific tasks. This review contributes valuable insights into Arabic NLP and enriches the existing body of knowledge.

# INTRODUCTION

With the advent of the Internet and the development of Web 2.0, various sources are producing large quantities of data, including social media users. This phenomenon has led

to the development of automatic text classification systems, single-label, and multi-label, to manage and utilize information efficiently (*Taha & Tiun, 2016*). While multi-label text classification has been extensively studied in high-resource languages like English, it remains underexplored for low-resource languages such as Arabic (*Al-Ayyoub et al., 2021*; *Elnagar, Al-Debsi & Einea, 2020*; *Qarqaz & Abdullah, 2021*). According to *Al-Salemi et al. (2019)*, *Elnagar, Al-Debsi & Einea (2020)* a significant issue with learning classifiers on multi-label datasets is a class imbalance (*Imran et al., 2022*), which affects the performance of the classification learning (*Taha et al., 2021*). Additionally, data adequacy is a persistent challenge affecting the models' training process (*AlAwawdeh & Abandah, 2021*; *Beseiso & Elmousalami, 2020*; *Talafha et al., 2019*).

To address these challenges, various data augmentation techniques have been employed for Arabic text classification (ATC). Data augmentation (DA) was initially developed for the computer vision domain (*Duwairi & Abushaqra, 2021*) as well as for speech recognition systems (*Lounnas, Lichouri & Abbas, 2022*) or in named entity recognition (*Sabty et al., 2021*). Nevertheless, implementing DA in natural language processing (NLP) is particularly challenging. In contrast with images or speech, textual data is not easily transformed without risking the loss of its semantic integrity. Various DA techniques have been applied to address this issue such as paraphrasing, noise injection, using pre-trained language models, generative models, back translation, and so on (*Li, Hou & Che, 2022*).

## Related works

This section will summarize the review and survey articles discussing data augmentation or text classification in Arabic or general (in other languages). *Feng et al. (2021)*, *Shorten, Khoshgoftaar & Furht (2021)*, *Bayer, Kaufhold & Reuter (2022)* introduced the concept of data augmentation to the research community. *Li, Hou & Che (2022)* discussed DA methods in three categories namely paraphrasing, noise, and sampling to enhance data training diversity. The study highlighted the applications of DA, tasks, and challenges. At the same time, *Bayer, Kaufhold & Reuter (2022)* provided a comprehensive taxonomy of data augmentation methods. Their review offers a concise overview by highlighting promising approaches and future directions. *Feng et al. (2021)* discussed data augmentation techniques and their application in NLP. The study summarized the trends and challenges associated with DA approaches. *Shorten, Khoshgoftaar & Furht (2021)* summarized the significant data augmentation motifs and frameworks for textual data. The authors addressed the challenges of overfitting and generalization in deep learning and compared the application of DA to computer vision.

Meanwhile, *Abdeen et al. (2019)* discussed Arabic text classification over two decades (2000–2019). They reported on the techniques used for text classification, the most popular datasets, feature selection methods, and the effect of stemming methods. *Aljedani, Alotaibi & Taileb (2020)* reviewed multi-label Arabic text classification techniques and related applications and challenges. *Alammary (2022)* introduced a survey of BERT models and their applications in ATC. They compared Arabic BERT to English BERT from the performance perspective, underscored the limitations, and suggested areas for future research. Next, *Alruily (2021)* comprehensively analyzed the classification of Arabic Tweets

which was carried out based on lexicon-based approaches, machine learning algorithms, as well as supervised, unsupervised, and hybrid learning. The review of *Wahdan, Al-Emran & Shaalan (2024)* focused on ATC, its area, application, and future research direction. The authors suggested different areas for exploration, such as handling unbalanced dataset issues and improving preprocessing techniques like stemming or tokenization.

## Motivation of the study

Despite the growing body of research on text classification and data augmentation within the NLP field, there remains a significant gap concerning ATC specifically enhanced through DA techniques, as shown in Table 1. This review addresses this critical gap, offering the first comprehensive review of data augmentation strategies tailored for ATC. Existing reviews and surveys have either focused on ATC without delving into data augmentation (*e.g.*, *Abdeen et al., 2019*; *Aljedani, Alotaibi & Taileb, 2020*; *Alruily, 2021*; *Alammary, 2022*; *Wahdan, Al-Emran & Shaalan, 2024*) or have explored data augmentation methods without specific attention to Arabic text (*e.g.*, *Feng et al., 2021*; *Shorten, Khoshgoftaar & Furht, 2021*; *Li, Hou & Che, 2022*; *Bayer, Kaufhold & Reuter, 2022*). Consequently, this review strives to contribute substantially to the research community by synthesizing these two vital areas, providing a foundational resource for future research and practical applications in Arabic NLP.

Therefore, this review article addresses several research questions regarding applying DA techniques in the context of ATC. It tries to answer the following questions:

RQ1: What are the DA techniques used for Arabic text classification?

RQ2: Does the augmented data get evaluated?

RQ3: What domains use the DA techniques in the context of the Arabic language?

RQ4: What is the importance of DA techniques in the context of the Arabic language?

RQ5: What are the challenges of using DA techniques in the Arabic language?

## The intended audience

Two distinct yet interconnected groups are the target audience for this research article. The first group is composed of academic researchers specifically those in the Arabic natural language processing (ANLP) domain who want to explore innovative DA techniques tailored to ATC and need to start or proceed with their work by exploring the latest techniques, what datasets, in what domains, and the challenges of data augmentation in text classification for the Arabic language. The second group is the computer science developers who are familiar with technical implementations but wish to better understand the specific challenges that arise in Arabic NLP and provide potential tools or already embedded methods to speed up the research for Arabic NLP. In addition to identifying challenges and future directions for Arabic NLP, this article highlights how these augmentation techniques can enhance the performance of text classification models. Furthermore, researchers and developers will be inspired to adopt these techniques and further advance ANLP.

The structure of this article is as follows: The next section explains the methodology used to conduct this review. It describes the inclusion and exclusion criteria with the

**Table 1 Comparison of different surveys in data augmentation and text classification.** This table presents a comparison of various surveys focusing on data augmentation techniques and text classification. The comparison highlights a gap in Arabic data augmentation, as none of the surveys on data augmentation considered the Arabic language. Similarly, the surveys or reviews on text classification did not explore the use of data augmentation methods.

| Ref | Objective | Language focus | DA | ATC |
|---|---|---|---|---|
| *Li, Hou & Che (2022)* | To introduce an overview of DA methods in NLP, application, and tasks. | The focus is on DA methods without concentrating on a specific language. However, the authors did not consider studies in Arabic text classification. | ✓ | × |
| *Bayer, Kaufhold & Reuter (2022)* | To present a concise review of DA approaches and future directions. | The focus is on DA methods without concentrating on a specific language. However, the authors did not consider studies in Arabic text classification. | ✓ | × |
| *Feng et al. (2021)* | To discuss trends of DA techniques and applications. | The focus is on DA methods without concentrating on a specific language. However, the authors did not consider studies in Arabic text classification. | ✓ | × |
| *Shorten, Khoshgoftaar & Furht (2021)* | To introduce DA motifs and compare them to computer vision techniques. | The focus is on DA methods without concentrating on a specific language. However, the authors did not consider studies in Arabic text classification. | ✓ | × |
| *Abdeen et al. (2019)* | To discuss Arabic text classification methods, dataset, stemming effects, and feature selection methods. | The focus is on Arabic text classification without considering DA. | × | ✓ |
| *Aljedani, Alotaibi & Taileb (2020)* | To review multi-label text classification in Arabic and its techniques. | The focus is on multi-label Arabic text classification without considering DA. | × | ✓ |
| *Alammary (2022)* | To introduce BERT models for Arabic text classification. | The focus is on Arabic text classification for the BERT model without considering DA. | × | ✓ |
| *Alruily (2021)* | To focus on Arabic tweet classification and its methods. | The focus is on classifying Arabic tweets without considering DA. | × | ✓ |
| *Wahdan, Al-Emran & Shaalan (2024)* | To focus on Arabic text classification, its area, application, and future research direction. | The focus is on Arabic text classification without considering DA. | × | ✓ |

**Note:**
DA, Data augmentation; ATC, Arabic text classification.

assessment of the selected articles. Followed by analysis and discussion of selected articles. The analysis highlights in depth the DA techniques, the importance, and the challenges. In the last section, we will conclude the article.

## SURVEY METHODOLOGY

This section highlights the methodology used for this review article to ensure a comprehensive approach. We examined different databases and developed different queries to obtain the relevant articles. In addition, we established a set of inclusion and exclusion criteria in selecting the most relevant articles on DA. This review applied a thorough analysis of the selected articles, providing insights into the currently used DA techniques as well as the gaps, importance, and challenges of these DA methods specifically for ATC.

### Search strategy

The search was conducted using academic databases, including Web of Science (WOS), ACM Digital Library (ACM), IEEE Xplore Digital Library, ACL Anthology, and

ScienceDirect (SD). Furthermore, a query was conducted on a connected article website to get the most relevant articles on Arabic data augmentation (ADA). The search was limited to materials published from 2019 to 2024 to obtain the most recent ones. Additionally, only journal articles and conference articles were selected. All the documents that can be accessed with their content were selected; otherwise, they were discarded. Meanwhile, different keywords were used including "data augmentation", "Arabic text", "paraphrasing", "data generation", and "text classification".

## Inclusion criteria

– Articles published between 2019 and 2024,
– Articles in English,
– Articles including journal and conference articles,
– Articles in full-text access,
– Articles address data augmentation in Arabic text classification.

## Exclusion criteria

– Articles address data augmentation in other languages,
– Articles address data augmentation in Arabic, but they are in image or voice,
– Articles cannot access their content.

To assess the collected articles, first, articles were filtered by screening their titles and abstract contents to ensure that only relevant articles were selected. The availability of data augmentation techniques was the primary factor for selection. Then, we focused on the data augmentation methods for the Arabic language in text domains including sentiment analysis, text detection, text identification, or wherever it contains textual datasets. All the information was recorded for further analysis. In the end, only 20 articles out of 287 articles were found to comply with the required criteria and passed the filtration process. Then these selected articles were analyzed from different angles including motivation, data augmentation technique, domain, dataset, and data source as follows:

- **Motivation**

Most researchers conducted a study due to a noticeable problem in previous work. The criterion for motivation refers to why these DA approaches were applied in the research.

- **Data augmentation techniques**

Several methods are used to perform the augmentation, ranging from the simplest ones like feeding the punctuation marks to the dataset, to the complicated ones using the generative models. This criterion refers to the DA techniques used for the Arabic language.

- **Domain**

Different applications and domains use data augmentation methods. This criterion refers to the most prominent fields employing DA methods in the Arabic context. These

domains may concentrate on text, speech, or images. However, the scope of this article is only on textual data for augmentation.

- **Datasets**

  Applying data augmentation in different domains leads to the usage of various datasets. This criterion clusters the datasets for DA approaches based on their characteristics or applications to highlight the benchmark dataset in Arabic.

- **Data source**

  Conducting data augmentation techniques on textual data triggers another criterion that points to the source of this data. This criterion will explain more about the data sources used in the selected articles such as social media, news, and reviews. Besides that, this criterion considers different varieties of Arabic, including Modern Standard Arabic (MSA) or different varieties.

## CHALLENGES OF ARABIC TEXT CLASSIFICATION

Arabic is the official language of 22 nations, with over 400 million speakers. It is the fifth most widely used language (*Alzuabidi et al., 2023*). There are three primary varieties of Arabic: i) Classic Arabic (CA), *i.e.*, an Arabic language employed in literary works and the Quran, ii) Modern Standard Arabic (MSA) which is used for writing, news, TV, and formal talks, and iii) Arabic dialect (AD) which is utilized in everyday conversation and casual exchanges (*Guellil et al., 2021*).

A significant advantage of ATC is that it provides valuable insights from a vast amount of textual information. Hence, it gives organizations or researchers a deeper understanding of public sentiment, identifies emerging trends, detects spam, and retrieves relevant information effectively or classifies data for some purposes. In addition, Arabic has a rich cultural and linguistic heritage, making it one of the most widely spoken languages in the world (*Hegazi et al., 2021*). Consequently, accurate ATC is crucial for language-specific applications or domains (*Alruily, 2021*). Despite this, ATC poses unique challenges that must be addressed to achieve accurate and reliable results.

- **Variations of Arabic:** The varieties of Arabic language dialects have made text classification challenging, as models trained in one dialect may not perform well in another. Also, mixing Modern Standard Arabic (MSA) with dialectal Arabic or non-Arabic words in social media texts introduces another challenge for classification tasks that must be handled first (*Alruily, 2021*). Additionally, as dialectal Arabic text uses the same Arabic characters, along with non-standard spelling, poor quality, and a common vocabulary, the identification problem of dialectal Arabic becomes even more complex (*Lulu & Elnagar, 2018*). The challenge with augmented data for dialects (Egyptian, Iraqi, Levantine, Gulf, or Maghrebi) is in ensuring that the generated text lies under the specific dialect, of which context must be considered.

- **Complicated morphology:** Unlike many languages, Arabic has an extensive vocabulary and complex morphology (*Farhan, Noah & Mohd, 2020*). It has a highly inflected

morphology, with roots forming words, prefixes, suffixes, and grammatical features determining word forms (*Abdelwahab, Al Moaiad & Bakar, 2023*). Due to this linguistic complexity, traditional classification algorithms face challenges in tokenizing, stemming, and identifying meaningful segments and patterns. For example, when tokenizing Arabic text, the algorithm may encounter words with different forms derived from the same root. These variations may include changes in prefixes or suffixes that alter the word's meaning. Traditional tokenization approaches might struggle to accurately identify and differentiate these variations, potentially leading to misinterpretations of semantics. Meanwhile, on the level of word augmentation, transforming the word by adding prefixes or suffixes on the original roots could be beneficial to augmenting the data. In addition, adding infixes that convey different meanings, like tenses of verbs or noun cases, is another way of augmenting data. On the other hand, incorporating these features to augment the data should have a more explicit process to help the model better understand semantic nuances associated with morphological structures.

- **Scarcity of available data:** Arabic text classification generally needs more availability of datasets (*Al-Salemi et al., 2019*). A relative lack of publicly labeled Arabic datasets makes developing, evaluating, and comparing classification models difficult (*Aljedani, Alotaibi & Taileb, 2020*). Methods like DA and transfer learning have been explored to overcome limitations imposed by limited training data. For example, paraphrasing, generative models, or back translation could effectively handle limited data. Applying these techniques could also improve the model's robustness and lead the model to generalize well on actual world data.

- **Noise and noise handling:** Arabic texts can contain various noise elements such as diacritics (Taksheel), abbreviations, misspellings, and informal language (*Hegazi et al., 2021*). Handling and pre-processing these noise elements can be crucial for improving classification performance. For instance, tackling the diacritics more efficiently will depend on the nature of the dataset, and the task of classification due to using them will change the sentence's meaning or grammar. For example, the words "ذَهَبَ" meaning "went" and "ذَهَبْ" meaning "gold" would affect the classification task since the former word does not identify any importance for the classification model. At the same time, the second word is crucial to classify the text under different labels like "economy", "international market", "women's jewelry", and so on.

## ANALYSIS

This section and the next one will go through the analysis of the selected articles and answer the research questions one by one.

### RQ1: Data augmentation techniques in Arabic text classification

The application of DA techniques in ATC has gained attention in recent years. Research on DA in other languages, such as English, has been more prevalent. Exploring DA specifically for Arabic text is a relatively newer area of study (*Duwairi & Abushaqra, 2021*). This attention to using DA is driven by the growing interest in NLP research and the need

to address the challenges of working with limited labeled or imbalanced data in Arabic. Figure 1 shows the overall trend of growing research interest and activity in the field over the past six years, despite some fluctuations. Different DA approaches have been applied to handwritten recognition, like in *Alwaqfi et al. (2023)*, *Eltay et al. (2022)*, *Lamtougui et al. (2023)* or speech recognition in *Al-Onazi et al. (2022)*, *Lounnas, Lichouri & Abbas (2022)* as is easier to implement state-of-the-art methods in these domains.

On the other hand, Arabic textual data has not been extensively studied, as we will clarify below. Arabic DA techniques are more commonly applied to image datasets than textual data in the context of the Arabic language. DA is widely used in Arabic literature primarily to categorize sentiment (*Refai, Abu-Soud & Abdel-Rahman, 2023*) in Arabic short texts. However, further research and exploration of DA techniques in ATC tasks is necessary, particularly in fields beyond sentiment analysis. Applying DA to textual data to handle class imbalance or data adequacy makes ATC models more accurate and perform better. DA techniques in Arabic are classified into data transformation, data generation, or expanded datasets using external data. Figure 2 shows the taxonomy of these methods which will be clarified in the following sub-sections.

### Data transformation

Data transformation (DT) includes techniques that modify text data to improve model performance. Data transformation may comprise paraphrasing, rule-based replacement, and back translation. These methods change the original data while maintaining integrity and meaning. Different DT techniques are used in ATC, such as paraphrasing using Arabic WordNet, word embedding, or the BERT model. Moreover, techniques like word shuffling, deletion, or swapping are used under noising methods. The back translation method is another type of DT method, which entails the translation of a text into another language and then back-translating it into the original language.

*Paraphrasing*

The term "paraphrase" is commonly used in natural language to describe an alternative way to convey the same information as its original form. Consequently, paraphrasing can increase the quantity of data available by generating them. Paraphrasing can be performed using synonym replacement, word embedding, or replacing the masked BERT tokens. *Duwairi & Abushaqra (2021)* applied synonym replacement using Arabic WordNet (*Elkateb et al., 2006*). By doing so, the authors paraphrase on the level of a word. Meanwhile, *Alkadri, Elkorany & Ahmed (2022)* used word embedding, specifically AraVec (*Soliman, Eissa & El-Beltagy, 2017*), as a substitution technique to enhance the diversity of the training dataset. They employed the 300-dimensional CBOW vectors of AraVec trained on Twitter, referred to as (AraVec-TWI) (*Soliman, Eissa & El-Beltagy, 2017*). The researchers identified non-stop words from tweets and then chose similar words from AraVec, applying a similarity threshold of 0.5 to ensure that the most comparable words were selected. As a result, the original tweets were replaced by similar words. The dataset was augmented with three examples for each tweet, which resulted in a doubling of spam

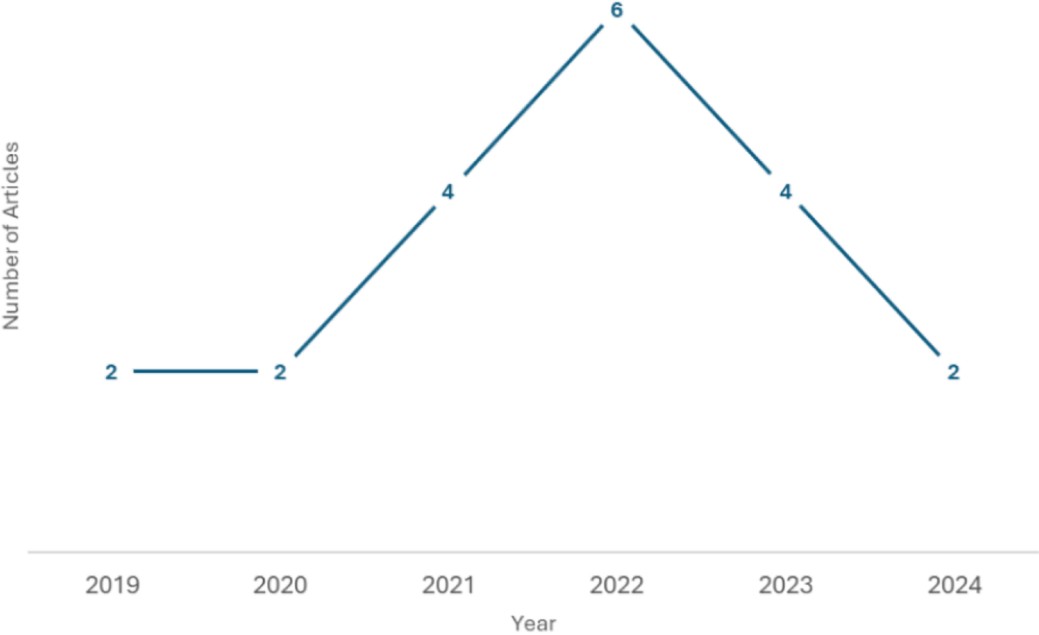

**Figure 1 Data augmentation articles distribution by year.** The figure displays the number of articles published each year, with data points marked along the trend line to indicate the exact count for each year.

tweets in their experimental setup. Table 2 shows an example of the paraphrasing technique to augment the data.

Meanwhile, *Fadel, Abulnaja & Saleh (2023)* proposed a DA method for Arabic Aspect-Based Sentiment Analysis using Fast Text and AraVec, along with the AraBART contextual language model. They generated dense vectors for unique aspects and sentiment terms, selected synonyms using cosine similarity, and replaced 30% of words in sentences. Besides, they used AraBERT to augment another dataset by masking one word at a time and predicting the masked word. However, they achieved the best results with Fasttext augmentation with increasing F1 score for aspect term extraction (ATE) from 75.94% to 78.56%, the F1 score for aspect polarity classification (APC) from 76.74% to 78.87%, and accuracy from 91.5% to 93.18%. Uniformaly, *Bensoltane & Zaki (2023)* applied a random masked method using the BERT model to augment the original data. The authors selected a random word from a news tweet and masked it. Next, they applied BERT to make predictions for the masked words since BERT considers the context of the entire sentence to predict possible values for the masked candidates. In this technique, the masking process is applied iteratively, with each successive masked word addressed.

*Back translation*
In addition to paraphrasing on the word level, *Bensoltane & Zaki (2023)* applied document-level augmentation using the back translation technique. The researchers translated the news posts and comments into other languages, including English, French,

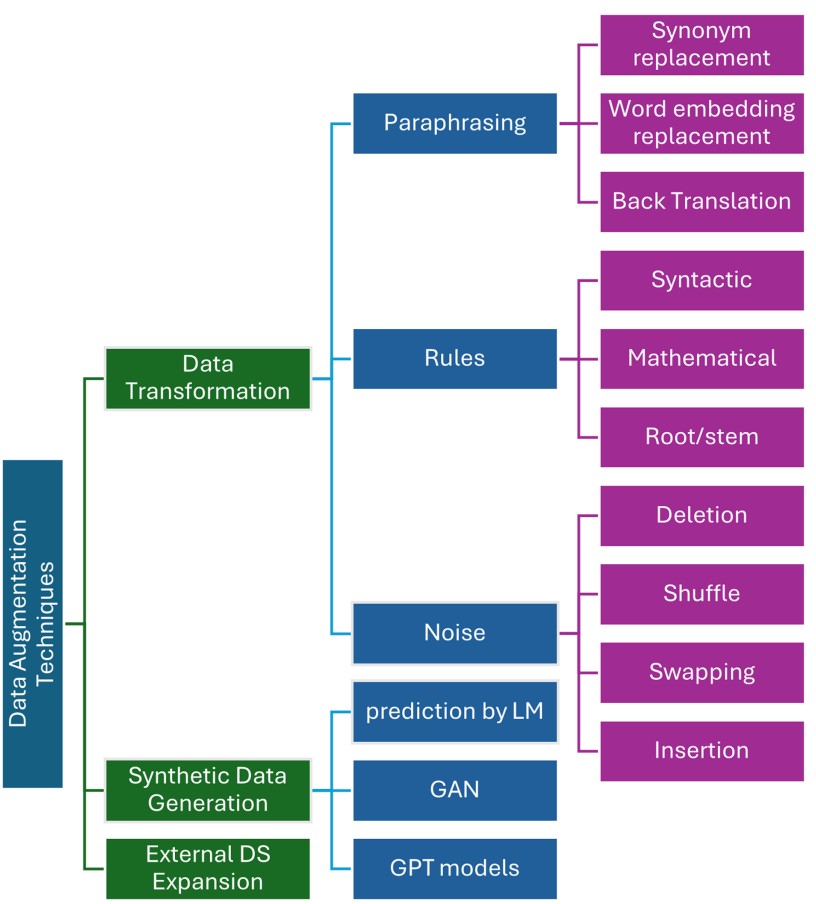

**Figure 2** Taxonomy of data augmentation techniques in Arabic text classification.

**Table 2 Paraphrasing example.** This table provides an example of paraphrasing as demonstrated in *Alkadri, Elkorany & Ahmed (2022)*. The paraphrased Arabic text shows changes in synonymous word choices (*e.g.,* "مضيئة" to "مشرقة" and "المظلم" to "المعتم"). However, the corresponding English translation retains the word "dark," highlighting a difference in lexical variations across languages.

| Original text | Augmented text |
| --- | --- |
| رائعة كنجمة مضيئة في سماء اليمن المظلم | جميلة كنجمة مشرقة في سماء اليمن المعتم |
| **Wonderful** as a shining star in the *dark* Yemeni sky | **Beautiful** as a bright star in the *dark* Yemeni sky |

Note:
2[nd] row is the translated English for the Arabic text, where the translated word "*dark*" is not changed for the translation version but is changed in Arabic. Words in bold represent the same paraphrasing, as do those underlined and italicized.

Dutch, and Spanish, and back-translated them into Arabic. When the translation is different, it is considered as new data.

*Rule-base*

*Duwairi & Abushaqra (2021)* applied grammatical, syntactic, and negation rules on the phrase level to augment their dataset and train machine learning models for sentiment analysis tasks on top of synonym replacement. *AlAwawdeh & Abandah (2021)* employed rule-based approaches to enlarge the dataset for Arabic question prediction tasks. The

authors used certain mathematical rules including reflexive, symmetry, and transitive relation applied to pairs of questions. Accordingly, they combined the generated dataset with the original, building a distinct dataset for each rule. Similarly, *Hamza et al. (2022)* employed the same rules (reflexive, symmetry, and transitive relation) to expand the duplicate and non-duplicate questions. Hence, the authors enhanced the performance of neural models for duplicate question detection tasks. While *Laskar et al. (2022)* employed the Arabic Light Stemmer-based root/stem substitution to augment the dataset for the propaganda detection task. Therefore, they increased the dataset from 504 to 1512. Then they used this augmented data to classify tweets in a multi-label classification task and a sequence tagging task to illustrate what kind of propaganda technique was used with span.

*Noising*

Noising-based approaches are slightly different from paraphrasing. The paraphrasing goal is to maintain the semantic similarity between the original and augmented text to preserve the meaning. On the other hand, the noising-based techniques inject a nuanced noise into the data, which does not significantly affect the semantics, producing slight variations from the original data. Noising-based methods represent swapping, shuffling, insertion, or deletion of words (*Li, Hou & Che, 2022*). Based on the word level augmentation, *Mohammed & Kora (2019)* utilized word shuffling to expand the dataset for training Arabic sentiment analysis. Within the small size context window, the authors randomly changed the order of words. Likewise, *Talafha et al. (2019)* applied a DA method to enlarge the MADAR dataset size. The authors employed word shuffling since they claimed that the order of words in a sentence was not essential. As a result, they generated five new sentences for each sentence by using different random seeds for each input sentence.

*Omran et al. (2023)* utilized random word swapping to handle the scarcity of training datasets. The authors randomly selected pairs of words within a sentence and exchanged their positions. This process was repeated N times to reach the desired result. Although this approach preserves the same features of the sentence semantically, it must be corrected. In addition, *Wafa'Q, Mustafa & Ali (2022)* employed word swapping and word deletion methods to augment the sarcastic class, representing less than half of the non-sarcastic class for the sarcasm detection task. In the word-swapping technique, the authors randomly changed the position of two words in a sentence. Random deletion involves randomly deleting words from a sentence based on predefined thresholds determined by uniformly generated numbers between 0 and 1. Using these two techniques, the authors improved the representation of the sarcastic class and mitigated the imbalance among the classes in the dataset. *Badri, Kboubi & Habacha Chaibi (2024)* used the word insertion from the NLPAug library using the AraBERT model to insert a new word into the dialect's datasets. Table 3 summarizes these techniques, including their specific methods and how they are implemented in ATC.

### Synthetic data generation

The generation process using generative models is another approach used to augment textual data. For instance, *Refai, Abu-Soud & Abdel-Rahman (2023)* applied the Arabic

**Table 3 Data transformation methods.** This table outlines various data transformation methods used in Arabic data augmentation. It distinguishes methods based on their types, including rule-based transformations, paraphrasing techniques, and noise injection methods. Relevant studies are cited for each approach.

| Method | Type of method | Ref. |
|---|---|---|
| Paraphrasing | Arabic WordNet | *Duwairi & Abushaqra (2021).* |
| | AraVec | *Alkadri, Elkorany & Ahmed (2022)* |
| | BERT | *Bensoltane & Zaki (2023)* |
| | Back translation | *Bensoltane & Zaki (2023)* |
| Noise | Shuffling | *Talafha et al. (2019)* \| *Mohammed & Kora (2019)* |
| | Swapping | *Wafa'Q, Mustafa & Ali (2022)*, *Omran et al. (2023)* |
| | Deletion | *Wafa'Q, Mustafa & Ali (2022)* |
| | Insertion | *Badri, Kboubi & Habacha Chaibi (2024)* |
| Rules | Syntactic or grammatical rules | *Duwairi & Abushaqra (2021)* |
| | Symmetric, transitive, and reflexive rules | *AlAwawdeh & Abandah (2021)* \| *Hamza et al. (2022)* |
| | Root/stem substitution | *Laskar et al. (2022)* |

GPT2 model, which trained on Arabic data from Wikipedia and other data from the Internet. The researchers utilized AraGPT2 (*Antoun, Baly & Hajj, 2020*) to augment the minority classes of their sentiment analysis datasets. They employed the AraGPT2 model to generate new data for the tweet/review input text fed to the model. Subsequently, they computed the similarity scores, Cosine, Jaccard, Euclidean, and Bleu, to verify that the generated text is related to the original, ensuring its qualification as augmented. They tested the performance of the augmented data using AraBERT to evaluate the influence of augmentation on the model's performance. Analogously (*Shah et al., 2024*) used the ChatGPT-3.5 generative model for DA on Unimodal (Textual) Propaganda Detection to handle the class imbalance problem. Although they addressed multimodal propaganda detection in three subtasks: text classification, image classification, and multimodal classification in their article, however, we included this study based on subtask 1 which is Unimodal (Textual) Propaganda Detection. They aimed to detect propaganda techniques with the corresponding span in a news paragraph or tweet. In addition, they fine-tuned different Arabic language models including but not limited to AraBERTv1, AraBERTv2, and AraGPT2, *etc.*, and they achieved 0.80 macro F1 on the augmented data.

Whereas *Beseiso & Elmousalami (2020)* followed the word level augmentation by training a language model on their dataset to forecast the last word in a tweet. New data was generated systematically using prediction words based on the associated probabilities. The authors aimed to increase the diversity of the dataset by training the language model on their existing dataset, thus generating multiple data points for each new sentence. This methodology calculated three probabilities for each new sample, resulting in three distinct samples per input. Based on the probabilistic nature of language models, this augmentation strategy provides a practical method for enhancing the training dataset and improving model performance. Meanwhile *Carrasco, Elnagar & Lataifeh (2021)* employed the modified SentiGAN to augment Iraqi, Levantine, Egyptian, Gulf, and Maghrebi

dialects to overcome the scarcity of annotated dialectal Arabic data. The authors used five generators and five discriminators to generate and assess each dialect. Unlike classical generalized adversarial networks (GAN), they utilized the penalty function using the Jaccard similarity score for the discriminator to ensure the novelty and diversity of the augmented data. Their method entailed using ordinary words to generate new data, reducing the vocabulary size. However, reducing the vocabulary size could cause a slight loss of richness and variety in the language.

On the other hand, *Gaanoun & Benelallam (2022)* used the idea of merging and then splitting they named it Mix to increase the number and diversity of the training dataset for the Propaganda Detection task. They split the original samples in Train and Dev datasets into individual records each representing a specific propaganda technique. Then they randomly mix these chunks for different techniques to generate synthetic new data. This approach produced a dataset with 2,000 synthetic samples that concatenated with the original dataset to expand the training dataset and achieved micro F1 0.455 compared to 0.43 micro F1 for the non-augmented dataset on the Dev set and 0.58 micro F1 on the Test set. Table 4 illustrates these methods, and the models used. To quickly see the development of DA in textual Arabic data, Table 5 presents the distribution of various DA methods applied in ATC from 2019 to 2024. It highlights the introduction and adoption of different techniques over the years. For instance, DA in ATC was first observed in this study scope in 2019 with noise methods such as swapping, deletion, or shuffling. However, this flow has been improved over the years by using sophisticated methods like paraphrasing with low context (word embedding) or high context (BERT models), using rules, or advanced generation models like GAN or generative pre-trained transformer (GPT).

### Data expansion using external dataset

Expansion of the training dataset with external data is another method for augmenting text used in Arabic NLP. For instance, *Gaanoun & Benelallam (2020)* augmented the original dataset of Arabic dialect identification using an external dataset named Unlabeled-10 M. They performed a pseudo-labeled using the initial model trained on the TRAIN dataset. High predictions based on softmax probabilities are filtered, retaining the top 99th percentile for majority countries and the 80th percentile for minority countries to address dialect imbalances. These high-confidence pseudo-labeled tweets are then combined with the original TRAIN dataset to create an augmented dataset, effectively expanding the training data. This augmented data is used iteratively to train new models, enhancing performance and generalization while leveraging external, previously unlabeled data. While *Abuzayed & Al-Khalifa (2021)* applied the DA by adding new documents to the dataset for the sarcasm detection and sentiment analysis tasks. They leveraged the sentiment analysis dataset ASAD based on their hypothesis that negative tweets are always sarcastic. Therefore, they replaced the negative labels with "True" indicating sarcasm, and positive labels with "False" (non-sarcasm). By using this method, they added 4,930 false tweets and 4,739 true tweets to the original dataset producing 22,217 tweets for the training set. For the sentiment analysis task, 4,930 positive tweets and 4,739 negative tweets from

**Table 4 Data generation methods.** This table highlights various data generation methods utilized in recent studies. It includes the corresponding types of models, such as language models, GANs, and hybrid approaches, with references to relevant studies for further review.

| Method | Type of model | Ref. |
|---|---|---|
| Generation | Prediction by LM | *Beseiso & Elmousalami (2020)* |
| | GAN | *Carrasco, Elnagar & Lataifeh (2021)* |
| | Synthetic generated by combination of existing chunks | *Gaanoun & Benelallam (2022)* |
| | AraGPT2 \| ChatGPT3.5 | *Refai, Abu-Soud & Abdel-Rahman (2023)* \| *Shah et al. (2024)* |

**Table 5 Data augmentation techniques in Arabic over the years.** This table provides an overview of the evolution of data augmentation techniques over the years (2019–2024). It highlights key methods, and their development timeline, showcasing the progress and trends in the field of Arabic data augmentation in text classification.

| DA methods | 2019 | 2020 | 2021 | 2022 | 2023 | 2024 |
|---|---|---|---|---|---|---|
| Paraphrasing (synonym replacement) | | | ✓ | | | |
| Paraphrasing (masked BERT) | | | | ✓ | | |
| Paraphrasing (word embedding) | | | | ✓ | | |
| Paraphrasing (back translation) | | | | | ✓ | |
| Noise | ✓ | | | ✓ | ✓ | ✓ |
| Rules | | | ✓ | ✓ | | |
| Prediction by LM | | | ✓ | | | |
| Generation by GAN | | | ✓ | | | |
| Generation by GPTs | | | | | ✓ | ✓ |
| Generation by combination of existing chunks (text) | | | | ✓ | | |
| Expansion with external dataset | | ✓ | ✓ | | | |

ASAD were added to the original dataset resulting in a new augmented dataset. They increased the F1 score by 15% for both tasks.

However, *Hussein et al. (2022)* handled label imbalance by merging the training, dev, and test datasets and then re-splitting them using multi-label stratification to achieve a more balanced label distribution for the propaganda detection task to identify the propaganda techniques and span. Although, underrepresented labels remained an issue, so they added a similar external dataset which is (SemEval-2020 Task 11) about propaganda detection in news articles translated from English to Arabic using RapidAPI. A mapping function aligned the labels between SemEval from English and WANLP to Arabic labels in this task. This method added 3,938 sentences, expanding the training data from 504 to 4,442 instances, which helped to address label imbalance, and they achieved a 0.649 micro F1 score.

## RQ 2: Evaluation of data augmentation in Arabic text classification

Assessing the effectiveness and impact of DA techniques in ATC is essential. This subsection focuses on analyzing the second research question of this article, which concentrates on evaluating the efficacy of these techniques by using specific metrics. The

evaluation involves measuring the impact on text quality (*Refai, Abu-Soud & Abdel-Rahman, 2023*), model accuracy (*Hamza et al., 2022*), and task-specific performance such as on the Arabic dialect (*Carrasco, Elnagar & Lataifeh, 2021*; *Talafha et al., 2019*). This subsection aims to provide insights and understand how DA techniques have been assessed for the reviewed articles.

❖ **Evaluating data generated by generation models**

Studies in this category first evaluate the quality of generated data using similarity scores before proceeding to classification. This approach ensures that the augmented data is relevant and maintains semantic integrity before it is used for training machine learning models.

**Similarity scores**

*Refai, Abu-Soud & Abdel-Rahman (2023)* used four similarity scores namely Cosine (*Zobel & Moffat, 1998*), Euclidean (*Danielsson, 1980*), Jaccard (*Ivchenko & Honov, 1998*), and BLEU (*Papineni et al., 2002*) methods to ensure the context, semantics, novelty, and diversity of the generated data. Likewise, *Carrasco, Elnagar & Lataifeh (2021)* used the discriminator of GAN, but the authors modified it to use the penalty function to ensure diverse instances using the Jaccard similarity score. These evaluations shed light on the ability of DA techniques to generate additional instances that effectively capture the nuances and complexities of Arabic text.

❖ **Evaluating data created by transformation methods or external dataset**

Studies employed transformation methods, such as synonym replacement or back translation, or augmented using external datasets directly used the augmented data for training machine or deep learning models. The evaluation focuses on the performance of these models in classifying the augmented data.

**Machine/deep learning metrics**

*Alkadri, Elkorany & Ahmed (2022)* applied three classifiers namely naïve Bayes (NB), linear regression, and linear SVM, before and after the augmentation of spam detection tasks to check the improvement in the classification process after using the DA methods. Meanwhile, *Beseiso & Elmousalami (2020)* focused on enhancing the sentiment analysis for short Arabic Twitter texts. The authors performed the augmentation to increase the size of the dataset, hence they evaluated the augmented data using sentiment analysis classifiers. *Mohammed & Kora (2019)* assessed the performance of long short-term memory (LSTM) using four measures namely average accuracy, average precision, average recall, and average F1-score. Likewise, *Omran et al. (2023)* used the LSTM classifier with augmented data to monitor the impact of augmentation on classifiers. *Wafa'Q, Mustafa & Ali (2022)* used the AraBERT and MARBERT transformer models to analyze the data, while *AlAwawdeh & Abandah (2021)* utilized a multinomial NB classifier to assess the augmented data. *Hamza et al. (2022)* evaluated the performance of DA on bidirectional LSTM. The proposed augmentation method by *Duwairi & Abushaqra (2021)* was evaluated for accuracy using three ML classifiers: SVM, NB, and k-nearest neighbors. *Abuzayed & Al-Khalifa (2021)* employed MARBERT and QCRI Arabic and Dialectal BERT (QARiB) to assess the augmented data using the accuracy, F1-score, precision and recall. *Gaanoun & Benelallam (2022)* used naïve Bayes and several Arabic BERT models to

evaluate the efficacy of augmented data through accuracy and macro F1 score. *Badri, Kboubi & Habacha Chaibi (2024)* utilized DziriBERT and BERT-based-Arabic models and got robust results for accuracy, F1-score, precision and recall. The AraBERT model is mostly used by several studies *Gaanoun & Benelallam (2022)*, *Hussein et al. (2022)*, *Laskar et al. (2022)*, *Fadel, Abulnaja & Saleh (2023)*, *Shah et al. (2024)* to investigate the effectiveness of augmented data in classification. Table 6 explains the evaluation type, methods, and metrics used to assess the augmented data.

## DISCUSSION

Twenty articles about data augmentation techniques for Arabic text classification were investigated. The following are discussions based on the comparison criteria previously developed.

### Motivation

We observed that the main goal of using DA methods is either to tackle data scarcity (*Mohammed & Kora, 2019*; *Talafha et al., 2019*; *Beseiso & Elmousalami, 2020*; *AlAwawdeh & Abandah, 2021*; *Carrasco, Elnagar & Lataifeh, 2021*; *Duwairi & Abushaqra, 2021*; *Hamza et al., 2022*; *Laskar et al., 2022*; *Gaanoun & Benelallam, 2022*; *Omran et al., 2023*; *Fadel, Abulnaja & Saleh, 2023*; *Badri, Kboubi & Habacha Chaibi, 2024*) or class imbalance (*Gaanoun & Benelallam, 2020*; *Abuzayed & Al-Khalifa, 2021*; *Wafa'Q, Mustafa & Ali, 2022*; *Alkadri, Elkorany & Ahmed, 2022*; *Hussein et al., 2022*; *Bensoltane & Zaki, 2023*; *Refai, Abu-Soud & Abdel-Rahman, 2023*; *Badri, Kboubi & Habacha Chaibi, 2024*; *Shah et al., 2024*) to improve the performance of the models. Eleven studies, as clarified in Table 7, were used to handle the limited availability of the labeled training datasets. In comparison, nine studies were used to tackle the class imbalance problem.

### Data augmentation methods

Different DA approaches created data to enhance the performance model. Some studies used more than one method for augmentation in case one is insufficient or for diversity. Hence, depending on the augmentation method used for this comparison criteria, each study was counted more than once. Two studies, *Bensoltane & Zaki (2023)*, *Beseiso & Elmousalami (2020)* used language model predictions to augment the dataset. In addition, one study *Bensoltane & Zaki (2023)* used the back translation method. Two studies *Alkadri, Elkorany & Ahmed (2022)*, *Fadel, Abulnaja & Saleh (2023)* used the word embedding to generate new data. Meanwhile, three studies *Carrasco, Elnagar & Lataifeh (2021)*, *Refai, Abu-Soud & Abdel-Rahman (2023)*, *Shah et al. (2024)* used generative models like GAN or GPTs models to perform the augmentation. Some studies focused on the linguistic characteristics of the language; methods like synonym replacement using Arabic WordNet or grammatical or syntactic rules and negation rules were applied (*Duwairi & Abushaqra, 2021*). The noise techniques such as random word swapping or shuffling (*Mohammed & Kora, 2019*; *Omran et al., 2023*), word deletion (*Wafa'Q, Mustafa & Ali, 2022*), and word insertion (*Badri, Kboubi & Habacha Chaibi, 2024*) were applied for the simplest augmentation. Furthermore, rule-based methods were employed by

**Table 6 Evaluation types with related evaluation techniques or methods and metrics.** This table categorizes various types of evaluation methods used for assessing the generated data or performance of text classification using the augmented datasets. Each evaluation type is paired with its associated methods/models and specific metrics that measure effectiveness or performance.

| Evaluation type | Evaluation techniques/Models | Metrics | Ref |
|---|---|---|---|
| Similarity scores | Cosine | Cosine similarity | *Refai, Abu-Soud & Abdel-Rahman (2023)* |
| | Jaccard | Jaccard index | *Refai, Abu-Soud & Abdel-Rahman (2023)*, *Carrasco, Elnagar & Lataifeh (2021)* |
| | Euclidean | Euclidean distance | *Refai, Abu-Soud & Abdel-Rahman (2023)* |
| | Bleu | Bleu score | *Refai, Abu-Soud & Abdel-Rahman (2023)* |
| Machine/Deep learning scores | Naïve Bayes (NB) | Accuracy, macro F1 | *Talafha et al. (2019)*, *AlAwawdeh & Abandah (2021)*, *Duwairi & Abushaqra (2021)*, *Alkadri, Elkorany & Ahmed (2022)*, *Gaanoun & Benelallam (2020)* |
| | KNN | Accuracy | *Duwairi & Abushaqra (2021)* |
| | SVM | Accuracy, macro F1 | *Duwairi & Abushaqra (2021)*, *Alkadri, Elkorany & Ahmed (2022)* |
| | Linear Regression (LR) | Accuracy, macro F1 | *Alkadri, Elkorany & Ahmed (2022)* |
| | LSTM | Accuracy, F1 | *Mohammed & Kora (2019)*, *Omran et al. (2023)* |
| | CNN | F1 | *Mohammed & Kora (2019)*, *Beseiso & Elmousalami (2020)* |
| | RCNN | F1 | *Mohammed & Kora (2019)* |
| | BiAttention BiLSTM | Accuracy, F1, AUC | *Hamza et al. (2022)* |
| | AraBERT | F1-score (macro) | *Refai, Abu-Soud & Abdel-Rahman (2023)*, *Wafa'Q, Mustafa & Ali (2022)*, *Badri, Kboubi & Habacha Chaibi (2024)*, *Fadel, Abulnaja & Saleh (2023)*, *Laskar et al. (2022)*, *Shah et al. (2024)*, *Gaanoun & Benelallam (2022)*, *Hussein et al. (2022)* |
| | MARBERT | Accuracy, F1, precision, recall | *Wafa'Q, Mustafa & Ali (2022)*, *Abuzayed & Al-Khalifa (2021)* |
| | DziriBERT, and Bert-base-Arabic models | Accuracy, F1, precision, recall | *Badri, Kboubi & Habacha Chaibi (2024)*, *Gaanoun & Benelallam (2020)* |
| | QARiB | Macro F1-score | *Abuzayed & Al-Khalifa (2021)* |

**Table 7 Motivation for applying data augmentation techniques.** This table presents the key motivations for applying data augmentation techniques, highlighting the primary reasons such as addressing data imbalance, and expanding dataset variability for more robust training outcomes.

| Motivation | Ref |
|---|---|
| Handle class imbalance | *Gaanoun & Benelallam (2020)*, *Abuzayed & Al-Khalifa (2021)*, *Wafa'Q, Mustafa & Ali (2022)*, *Alkadri, Elkorany & Ahmed (2022)*, *Hussein et al. (2022)*, *Bensoltane & Zaki (2023)*, *Refai, Abu-Soud & Abdel-Rahman (2023)*, *Badri, Kboubi & Habacha Chaibi (2024)*, *Shah et al. (2024)* |
| Overcome data scarcity | *Mohammed & Kora (2019)*, *Talafha et al. (2019)*, *Beseiso & Elmousalami (2020)*, *AlAwawdeh & Abandah (2021)*, *Carrasco, Elnagar & Lataifeh (2021)*, *Duwairi & Abushaqra (2021)*, *Laskar et al. (2022)*, *Hamza et al. (2022)*, *Gaanoun & Benelallam (2022)*, *Omran et al. (2023)*, *Fadel, Abulnaja & Saleh (2023)*, *Badri, Kboubi & Habacha Chaibi (2024)*. |

*AlAwawdeh & Abandah (2021)*, *Hamza et al. (2022)* using transitive, symmetry, and reflexive relations to augment the dataset. *Laskar et al. (2022)* used the root/stem rules using an Arabic light-stemmer to produce new augmented data. Moreover, some studies

*Gaanoun & Benelallam (2020)*, *Abuzayed & Al-Khalifa (2021)*, *Hussein et al. (2022)* augmented their original data by adding a new external dataset. Table 8 provides a summary of data augmentation techniques in Arabic text classification.

As we observed from Table 8, methods focusing on semantic preservation, such as paraphrasing *via* word embeddings or back translation, illustrate significant improvements in classification performance, particularly F1 scores. For instance, replacing similar words using word embeddings increased the F1 score from 65% to 89%. However, the dependency on pre-trained models and their inherent limitations, such as vocabulary constraints, points to a need for more adaptable and comprehensive embedding resources tailored for the Arabic language.

Conversely, noise-based techniques, such as random word swap or deletion, yielded inconsistent results. While these methods can introduce helpful variability, their unpredictable impact on semantic integrity suggests potential barriers. Random swapping achieved a moderate accuracy increase of 14.06% in some models, but often compromised the meaning of text, particularly for longer sentences. This suggests that noise-based augmentations need to be applied cautiously, potentially in combination with more sophisticated semantic-preserving methods to balance variability with coherence. The rule-based augmentation methods, mainly those employing syntactic and semantic rules, demonstrated considerable promise, with the highest reported accuracy of 93.86% using syntactic transformations like transitive or symmetric rules. This points out that rule-based methods can effectively capture linguistic transformations, offering more control compared to stochastic approaches. However, these approaches are labor-intensive and challenging to scale, as developing exhaustive linguistic rules is time-consuming.

The generation-based techniques, including those leveraging GANs, AraGPT2, and ChatGPT, showed significant improvements, with reported increases in F1 scores by up to 13% in certain datasets compared to producing synthetic data based on Mix method (split and merge) as illustrated in Table 8. Such methods create highly diverse and contextually relevant text, addressing the scarcity of labeled data. However, the computational cost and complexity of tuning these models are notable challenges. Despite these constraints, generation methods are powerful because they can synthesize realistic examples that enrich the training set far beyond what simpler strategies can achieve. Expanding data through external datasets also appears strategic to address the limitations of small native datasets in Arabic NLP. The integration of translated datasets, such as SemEval-2020, yielded a moderate boost to the performance of models like AraBERT, with an F1 score of 0.649. This highlights the benefits of cross-linguistic resource expansion, though translation quality and cultural context must be rigorously managed to avoid biases and inaccuracies. Consequently, a hybrid DA strategy for ATC is recommended to maximize effectiveness while balancing resource constraints and linguistic integrity. Despite the observations presented in this table, it is not possible to draw a definitive conclusion. It is important to note that the studies included in this article often do not use the same datasets or apply the same methods. Although in some cases the dataset is the same, the DA approaches vary. As a result, we cannot say with certainty whether this approach is suitable in all contexts.

**Table 8 Data augmentation techniques in Arabic text classification.** This table outlines various data augmentation (DA) techniques applied in Arabic text classification. It details each technique's advantages (pros), limitations (cons), the type of database or model it is associated with, improvements achieved, and relevant references.

| DA | Type of DA | Pros | Cons | Database/model | Improvement | Ref |
|---|---|---|---|---|---|---|
| Paraphrasing | Replacing synonyms using WordNet. | It is easy to implement and | Affected by the limited WordNet vocabulary. | Arabic WordNet | 42% increase in Acc. | *Duwairi & Abushaqra (2021)* |
| | Replacing similar words using a word embedding. | It utilizes pre-trained word embeddings for contextual similarity | Limited to the vocabulary of the embeddings. | AraVec \| Fasttext | Increases F1 from 65% to 89%. \| from 76.74% to 78.87%, | *Alkadri, Elkorany & Ahmed (2022)* \| *Fadel, Abulnaja & Saleh (2023)* |
| | Replacing masked tokens. | Uses contextual pre-trained models to create accurate data. | The created data's quality depends on the pre-trained model's training data and coverage. | BERT | Improves F1 from 37% to 50% for parties' class | *Bensoltane & Zaki (2023)* |
| | Paraphrasing the text using back translation methods. | It maintains semantic meaning while introducing syntactic diversity | Requires high-quality translation models and does not permanently preserve the original meaning perfectly. | Back-translation | Achieve 78.25% F1 for parties' class. Improving by 3%. | *Bensoltane & Zaki (2023)* |
| Noise | Swapping two words' positions. | Simple and easy to implement. | It can compromise the original meaning of the text. | Random swap | Improving Acc by 14.06%, 12.57% for LSTM on Bahraini, and MSA. MARBERT achieved an F1 of 52% and Acc of 78%. | *Omran et al. (2023)* \| *Wafa'Q, Mustafa & Ali (2022)* |
| | Shuffling the order of words inside a sentence. | It is simple and effective for short texts. | Can compromise the original meaning of the text and is less effective for longer texts. | Random shuffle | Achieved 67.51% F1 and Acc of 81.31%, increased by 8.3% for LSTM | *Talafha et al. (2019)* \| *Mohammed & Kora (2019)* |
| | Insert new word randomly. | It is easy to implement. | Can compromise the original meaning. | Random insertion using NLPAug library with AraBERT | Increased performance between 15–21% in Acc, F1, R compared to non-augmented data | *Badri, Kboubi & Habacha Chaibi (2024)* |
| | Deleting some words from the text. | It is simple to implement. | Can remove important information | Random delete | MARBERT achieved 52% F1 and 78% Acc. | *Wafa'Q, Mustafa & Ali (2022)* |
| Generation | Predicted new words using language models. | It generates alternatives relevant to the context and can handle rare or unseen words. | It is intensive in computation. Its performance is dependent upon the LM quality training dataset. | BiGRU trained on LM | F1 increased by 9.3%. | *Beseiso & Elmousalami (2020)* |
| | Generate new text/document. | It produces highly diverse and realistic text. | It requires high computational cost and needs careful tuning to avoid mode collapse. | GAN \| AraGPT2 \| ChatGPT3.5-4 | It increased Acc by 8%, \| and F1 by 13% in MOVIE 9% in ATT, 6% in ASTD, 4% in AraSarcasm \| achieved macro F1 0.80 | *Carrasco, Elnagar & Lataifeh (2021)* \| *Refai, Abu-Soud & Abdel-Rahman (2023)* \| *Shah et al. (2024)* |
| | Synthetic data generated by combination of existing chunks of text. | It controls the new synthetic data since they are still from the same dataset | It is time consuming specifically when the dataset size is big. | Split text to chunks then Mix | Increased the micro F1 from 0.43 to 0.455 | *Gaanoun & Benelallam (2022)* |

| DA | Type of DA | Pros | Cons | Database/model | Improvement | Ref |
|---|---|---|---|---|---|---|
| Expansion with external dataset | The pseudo-labelled technique was performed using unlabeled set and select data with higher SoftMax probability. | It added new features to the dataset. | It needs more caution and constraints on the unlabeled data to ensure it aligns with original labels of the dataset. | SoftMax probability. | AraBERT achieved 23.26% F1-score on country-level detection | *Gaanoun & Benelallam (2020)* |
| | Added new sentiment analysis dataset to the training based on the assumption that negative sentiment text always sarcastic and *vice versa* | It added new semantic features to the dataset. | May lead to loss nuances between negative and sarcastic since not always the negative sentiment is sarcastic and *vice versa*. | – | MARBERT achieved 0.80 F1 on sarcasm detection and 0.86 on sentiment analysis | *Abuzayed & Al-Khalifa (2021)* |
| | Add SemEval-2020 Task 11 English dataset after translating it to Arabic and mapping the labels to fit the current task | It enriches the diversity of the dataset. | The risk of adding translation errors and mismatch labels that affect model accuracy. | – | AraBERT achieved micro F1 score of 0.649 | *Hussein et al. (2022)* |
| Rules | Syntactic, grammar, and negation rules to augment data. | It provides precise control over the types of transformations applied. | It cannot handle ambiguous linguistic constructions effectively and is time-consuming. | User-defined rules | 42% increase in Acc. | *Duwairi & Abushaqra (2021)* |
| | Transitive, symmetric, and reflexive rules | It enhances the ability to identify similar questions. | The incorrect similarity assessments can cause errors. | Transitive, reflexive, and symmetry relations | Achieved 93.86% Acc, 93.37% macro F1, \| 93.84% weighted F1 | *AlAwawdeh & Abandah (2021), Hamza et al. (2022)* |
| | Root/stem substitution | It is simple to implement. | The effect of changing the grammar and semantics of text | Arabic light-stemmer | Achieved micro F1 0.602 | *Laskar et al. (2022)* |

**Note:**
All the results for the column "Improvement" illustrate the improvement after using the augmented dataset, so the comparison here is regarding applying the augmentation. Acc, Accuracy; F1, F1 score; R, Recall.

## Augmentation level

Most paraphrasing methods are applied to word-level augmentation since they are effective for altering a specific word to maintain the context and semantics of the text. Meanwhile, phrase or sentence-level augmentation is used mostly with rules to change more than one word. This augmentation shows significant improvement and is adequate for logical consistency and structure reasoning tasks. The Document-level augmentation using generation models or external datasets is highly effective in maintaining semantic coherence and augmenting diverse and contextual texts. However, it is essential to ensure the quality of the generated or added text and its alignment with the labels of the original dataset. Table 9 outlines the suitable level of augmentation for each DA technique.

## RQ3: Domain

Figure 3 categorizes the DA methods on various application areas such as sentiment analysis (*Mohammed & Kora, 2019*; *Beseiso & Elmousalami, 2020*; *Duwairi & Abushaqra, 2021*; *Abuzayed & Al-Khalifa, 2021*; *Refai, Abu-Soud & Abdel-Rahman, 2023*; *Omran et al., 2023*), dialect identification (*Carrasco, Elnagar & Lataifeh, 2021*; *Talafha et al., 2019*; *Gaanoun & Benelallam, 2020*), sarcasm detection (*Abuzayed & Al-Khalifa, 2021*;

**Table 9 Level augmentation for various data augmentation methods.** This table compares different data augmentation (DA) methods based on their augmentation levels, providing corresponding references for each method. It highlights the extent to which each method impacts data, ranging from word-level to document-level augmentation.

| DA method | Augmentation level | Ref |
|---|---|---|
| Paraphrasing (synonym replacement WordNet) | Word level | *Duwairi & Abushaqra (2021)* |
| Paraphrasing (masked BERT) | Word level | *Bensoltane & Zaki (2023)* |
| Paraphrasing (word embedding) | Word level | *Alkadri, Elkorany & Ahmed (2022)*, *Fadel, Abulnaja & Saleh (2023)* |
| Paraphrasing (Back translation) | Document level | *Bensoltane & Zaki (2023)* |
| Noise | Word level | *Wafa'Q, Mustafa & Ali (2022)*, *Badri, Kboubi & Habacha Chaibi (2024)*, *Mohammed & Kora (2019)*, *Omran et al. (2023)*, *Talafha et al. (2019)* |
| Rules | Word level | *Laskar et al. (2022)* |
| | Phrase/ Sentence level | *AlAwawdeh & Abandah (2021)*, *Hamza et al. (2022)* |
| Generation by prediction of LM | Word level | *Beseiso & Elmousalami (2020)* |
| Generation by GAN, AraGPT2, ChatGPT | Document level | *Carrasco, Elnagar & Lataifeh (2021)*, *Refai, Abu-Soud & Abdel-Rahman (2023)*, *Shah et al. (2024)* |
| Adding external data | Document level | *Abuzayed & Al-Khalifa (2021)*, *Gaanoun & Benelallam (2020)*, *Hussein et al. (2022)* |
| Mix "merge dataset and split to chunks" | Phrase/ Sentence level | *Gaanoun & Benelallam (2022)* |

*Wafa'Q, Mustafa & Ali, 2022*), question-answering tasks (*AlAwawdeh & Abandah, 2021*; *Hamza et al., 2022*), hate speech detection (*Badri, Kboubi & Habacha Chaibi, 2024*), spam detection (*Alkadri, Elkorany & Ahmed, 2022*) and propaganda detection (*Gaanoun & Benelallam, 2022*; *Hussein et al., 2022*; *Laskar et al., 2022*; *Shah et al., 2024*) and finally for aspect-based sentiment analysis (*Bensoltane & Zaki, 2023*; *Fadel, Abulnaja & Saleh, 2023*). Each domain's prominence indicates where DA techniques are most actively applied and explored, underscoring the interdisciplinary nature of NLP research in ATC.

Data augmentation techniques have been employed across various domains to enhance the performance of the classification models. In sentiment analysis, multiple methods such as AraGPT2, pre-trained language model predictions, and syntactic rules with WordNet have enriched the dataset and improved model accuracy and robustness. For instance, AraGPT2 has been applied to generate new texts, significantly enhancing F1 scores across various datasets. Similarly, pre-trained language model predictions and the use of WordNet for synonym replacement, combined with syntactic rules, have shown substantial improvements in sentiment classification tasks. Methods like shuffling word order and using generative adversarial networks (GANs) have been proven effective in dialect identification. GANs introduce variability and increase the robustness of models to handle the intricacies of different dialects. For question prediction and detection, rule-based approaches have achieved high accuracy and F1 scores, demonstrating the effectiveness of structured augmentation in this domain.

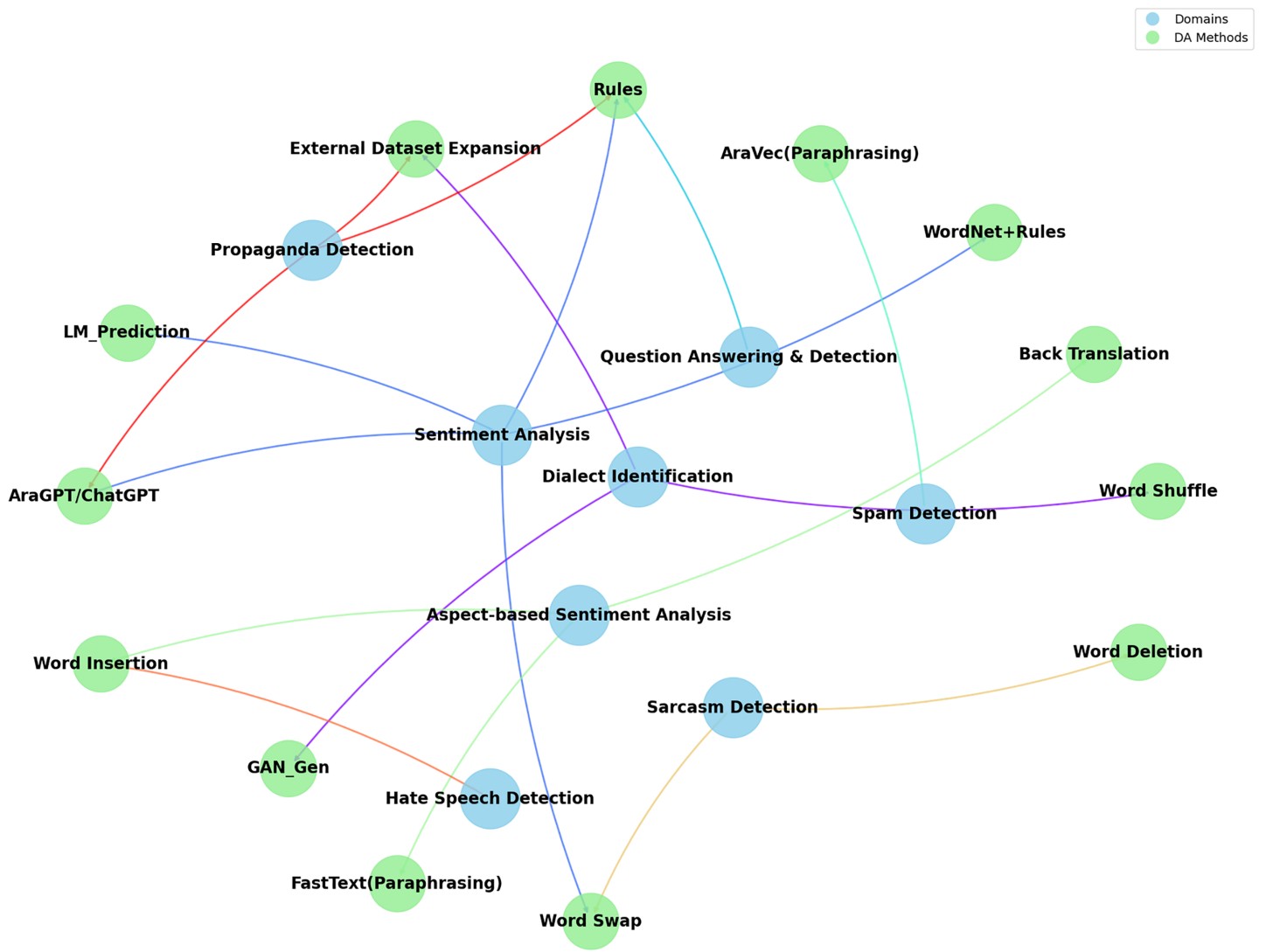

**Figure 3 Data augmentation methods per domains.**

The influence of data augmentation methods varies by domain, reflecting their potential suitability and impact. For instance, in sentiment analysis, techniques such as text generation have been suggested to help models generalize better by exposing them to a broader range of linguistic variations and expressions of sentiment. Similarly, GANs have been explored in dialect identification as they can simulate the structural patterns of words and phrases in different dialects. On the other hand, rules-based methods have been utilized in question prediction and detection to ensure that augmented data maintains logical consistency and relevance. However, it is essential to note that the limited number of studies addressing each of these applications makes it difficult to draw definitive conclusions about the most suitable data augmentation techniques for each domain. A more comprehensive examination of DA techniques across a broad range of studies is needed to validate their effectiveness for specific applications.

## Dataset

The availability of benchmark datasets is rare when it comes to the Arabic language. As shown in Table 10, two studies *Carrasco, Elnagar & Lataifeh (2021)*, *Talafha et al. (2019)* evaluated their methods on the same dataset, namely the Multi-Arabic Dialect Applications and Resources (MADAR) (*Bouamor et al., 2018*) dataset. This dataset contains 52,000 parallel sentences written in MSA and the local dialect of 25 cities. Besides that, two studies *AlAwawdeh & Abandah (2021)*, *Hamza et al. (2022)* evaluated their rules on the same dataset, *i.e.*, Mawdoo3. Sentiment analysis datasets mainly concentrate on tweets, product reviews, and hotel review data. Some studies used the Arabic Sentiment Tweets Dataset (ASTD) (*Nabil, Aly & Atiya, 2015*) for sentiment analysis (*Beseiso & Elmousalami, 2020*; *Refai, Abu-Soud & Abdel-Rahman, 2023*) as a benchmark dataset to conduct their experiments. Details of the datasets are explained clearly in Table 10.

Based on the dataset sizes presented in the table, there is a significant variation across Arabic NLP tasks and domains. A dataset such as Multi Arabic Dialect Applications and Resources (MADAR), used for dialect identification, contains more than 52,000 samples, while others have substantially fewer. In the case of product reviews, for example, only 300 reviews are included in the sentiment analysis dataset. As a result of this disparity in dataset sizes, DA techniques have become imperative, particularly in domains with limited resources or a lack of high-quality annotations. There is a complex relationship between dataset size and domain, and it varies depending on the task at hand. There are many types of sentiment analysis datasets, such as reviews and tweets, which range in size from just 300 to 63,000 tweets (LABRv2) (*Aly & Atiya, 2013*). The variability in data availability suggests that certain domains, like social media, are more easily accessed, whereas others, like product reviews, face more challenges during data collection and annotation. The necessity of DA becomes especially evident in domains with limited data, since models developed from small datasets may not be generalized in real-world situations.

## Data source

Table 10 exhibits a range of data sources, including curated datasets like Quora and iSarcasmEval (*Farha et al., 2022*), social media data like Facebook and Twitter, and product reviews from e-commerce sites like Amazon and souq.com. This range offers a good foundation for using data augmentation methods such as noise injection, paraphrasing, and back translation. For example, augmentation techniques like back translation might be helpful when product reviews were translated from English into Modern Standard Arabic (MSA) and then Bahraini. By using this technique, any potential translation-induced biases can be identified and mitigated by translating the content back into its native language. Additionally, it increases the dataset's linguistic diversity, which raises the accuracy of the models that were trained on it.

Furthermore, augmentation methods can help handle informal language, and dialects in social media data, which is a significant element in the datasets (*e.g.*, LABRv2, AraSarcasm (*Farha, Zaghouani & Magdy, 2021*)). Using strategies like paraphrasing or word substitution can create diverse data making the models more robust to different variations of the same sentiment. Augmentation confirms that data covers several linguistic

**Table 10 Dataset summary with their domains.** This table summarizes various datasets used in Arabic data augmentation for text classification, specifying their domains, dataset names, availability, dataset sizes, and data sources. References are provided for each dataset to facilitate further expansion.

| Ref | Domain | Dataset | Availability | DS size | Data source |
|---|---|---|---|---|---|
| *Carrasco, Elnagar & Lataifeh (2021)* and *Talafha et al. (2019)* | Dialect identification | MADAR | Available at: https://sites.google.com/nyu.edu/madar/ | 52 k MSA and dialect of 25 cities. | MSA + Dialects |
| *Beseiso & Elmousalami (2020)*, *Refai, Abu-Soud & Abdel-Rahman (2023)* | Sentiment analysis | ASTD | Available at: https://github.com/mahmoudnabil/ASTD | 10 k tweets for four classes were used by *Beseiso & Elmousalami (2020)*, while just 3,224 tweets were used by *Refai, Abu-Soud & Abdel-Rahman (2023)*. | Social media |
| *Shah et al. (2024)* | Propaganda detection | ArAIEval 2024 dataset | Available at: https://araieval.gitlab.io/task1/ | The dataset consists of 2,143 train, 312 validation, and 607 test pairs of text and images. | Tweet and news paragraphs |
| *Gaanoun & Benelallam (2022)*, *Hussein et al. (2022)* and *Laskar et al. (2022)* | Propaganda detection | WANLP 2022 dataset | Available at: https://gitlab.com/araieval/propaganda-detection | Train set: 504, and Dev set: 52 | Tweets |
| *AlAwawdeh & Abandah (2021)* and *Hamza et al. (2022)* | Question prediction and detection | Mawdoo3 | Not specified | 11,997 question pairs, in which 45% of these pairs' questions are duplicated, whereas the rest are not. | MSA |
| *Beseiso & Elmousalami (2020)* | Sentiment analysis | LABRv2 | Available at: https://paperswithcode.com/dataset/labr | 63 k tweets with three classes (POS, NEG, NEU). | It is not stated exactly, but it is social media data. |
| *Duwairi & Abushaqra (2021)* | Sentiment analysis | Product reviews | Available at: https://peerj.com/articles/cs-469/#supplementary-material | 300 reviews were collected from souq.com. | MSA |
| *Refai, Abu-Soud & Abdel-Rahman (2023)* | Sentiment analysis | AraSarcas, ASTD, ATT, and MOVIE | Available at: https://github.com/iabufarha/ArSarcasm-v2. https://github.com/hadyelsahar/large-arabic-sentiment-analysis-resouces. | 10,547 tweets, where 16% of them were snarky, 3,224 tweets with four classes, 2,154 reviews with two classes, and 1,524 reviews with three classes for each dataset, respectively. The classes are mostly POS, NEG, NEU, or OBJ. | Mixed |
| *Alkadri, Elkorany & Ahmed (2022)* | Spam detection | Self-collected data from Twitter. | Available upon request from authors. | 313 k tweets after filtration, 5 k of them annotated manually for spam/not spam. | It is not stated exactly, but it is social media data. |
| *Bensoltane & Zaki (2023)* | Aspect-based sentiment analysis (ABSA) | News posts and comments about the Gaza attacks in 2014 from Facebook. | Available from *Al-Ayyoub et al. (2017)* | The news posts train set is 1,811, and test is 454. The comments train is 10,902 and test set is 2,726. | MSA |
| *Mohammed & Kora (2019)* | Sentiment analysis | Self-collected from Twitter. | Available at: https://dataverse.harvard.edu/dataset.xhtml?persistentId=doi:10.7910/DVN/LBXV9O | 40 k tweets annotated manually with POS, NEG. Mixed with MSA and Egyptian dialect. | MSA + Egyptian |

| Ref | Domain | Dataset | Availability | DS size | Data source |
|---|---|---|---|---|---|
| *Omran et al. (2023)* | Sentiment analysis | Reviews of Amazon products in English, which are translated into MSA, and then into Bahraini. | Not specified | 5 k reviews in MSA that were translated from English using Google translator, then translated into Bahraini dialect by humans. | MSA + Bahraini |
| *Wafa'Q, Mustafa & Ali (2022)* | Sarcasm detection | iSarcasmEval dataset (*Farha et al., 2022*). | Available at: https://github.com/iabufarha/iSarcasmEval | 3,102 tweets | MSA + Nile, gulf, levantine, and maghrebi |
| *Hamza et al. (2022)* | Question detection | Quora | Not specified | 40 k from the Quora website translated into English using Google API, with a binary label indicating if the question is duplicated or not. | MSA |
| *Gaanoun & Benelallam (2020)* | Dialect identification | NADI dataset | Available at: https://nadi.dlnlp.ai/ | Train: 21,000, DEV: 4,957, and Test: 5,000. | Tweets |
| *Abuzayed & Al-Khalifa (2021)* | Sarcasm and sentiment detection | ArSarcasm-v2 | Available at: https://iabufarha.github.io/resources/ | Training is 12,548 tweets | Tweets |
| *Fadel, Abulnaja & Saleh (2023)* | Aspect-based sentiment analysis (ABSA) | SemEval 2016 task 5 (Arabic Hotels' reviews) | Available at: https://alt.qcri.org/semeval2016/index.php?id=tasks | Training set is 4,082 sentences and Testing set is 1,227 sentences | Mix |
| *Badri, Kboubi & Habacha Chaibi (2024)* | Hate speech recognition | Mono dialect (Tunisian, Egyptian, Lebanese), one multidialectal (Tun-EL). | Available at: https://github.com/NabilBADRI/Towards-Automatic-Detection-of-Inappropriate-Content-in-Multi-dialectic-Arabic-Text | Tunisian: 6,460, Egyptian: 1,100 Lebanese: 9,196, and Tun-EL: 18,291 | Dialect |

**Note:**
LABR, Large-scale Arabic Book Review.

expressions which are crucial for domains like sentiment analysis and sarcasm detection. Eventually, applying data augmentation techniques to these different data sources helps improve model performance by reducing biases and increasing the generalizability of the training data across various domains and dialects.

## RQ4: Importance of data augmentation in Arabic text classification

The scarcity of labeled datasets or imbalanced classes for ATC poses significant challenges. DA techniques offer a solution by artificially expanding and diversifying the dataset. In this context, data augmentation plays a crucial role in improving the performance of classification models. It enables the modeling of a more prominent and representative dataset, ultimately enhancing the effectiveness of ATC. In the following points, we will further explore data augmentation's specific benefits and advantages to ATC.

- **Addressing data scarcity:** This review article provides a comprehensive overview of different data augmentation (DA) techniques used to manage data scarcity in low-resource languages like Arabic. It examines the most effective DA techniques to overcome the limitations of training data required for classification models by

considering methods such as paraphrasing, noise addition, and data generation. For example, as mentioned earlier in the RQ1 section, *Carrasco, Elnagar & Lataifeh (2021)* utilized GANs to generate robust text, thereby overcoming data scarcity and improving classification models.

- **Increasing training data:** This article also explores how DA can enrich training datasets. It extends this line of research by examining the implementation of various DA techniques to augment training data. For instance, *Beseiso & Elmousalami (2020)* utilized a language model (LM) trained on 70% of their dataset to predict the last word and expand the dataset. This approach ensures augmentation with contextually appropriate data. They introduced diverse features into the dataset by selecting the top three predictions of the language model as a paraphrasing approach. For example, with the input sentence "I have to go to", the LM might predict "school, market, cinema" thereby enriching the dataset with new features. Similarly, using masked BERT, *Bensoltane & Zaki (2023)* created contextual semantic data which is crucial for making diverse training datasets and enhancing model performance.

- **Mitigating imbalanced class problems:** This article contributes to understanding class imbalance in classification datasets by discussing various DA methods to address this issue. For instance, *Refai, Abu-Soud & Abdel-Rahman (2023)*, *Shah et al. (2024)* examined the use of AraGPT2 or ChatGPT to generate new data for classes with minority samples. By expanding these classes, the authors balanced the overall dataset. The GPT model leverages powerful pre-trained models on extensive data to produce effective results, thereby addressing class imbalance.

## RQ5: Challenges and future work

Data augmentation is instrumental in enhancing the data pool for ATC. However, as transformative as these practices are, they face specific challenges due to the distinctive characteristics of the Arabic language. Linguistic complexities, multiple dialects, limited resources, and data imbalance weigh heavily on the efficacy of these techniques. The following points will delve into these challenges to advance the understanding and effectiveness of data augmentation in ATC.

- **Lack of research:** Data augmentation research for the Arabic context is limited compared to English. Future research should focus on comprehensive studies covering various Arabic text types. It may also be possible to identify the most promising DA techniques for Arabic text in future research by comparing their effectiveness and whether there is a difference between applying them to short text or long text, or MSA or dialect, as well as what DA methods are effective for each type of text. Therefore, researchers can provide more tailored solutions for Arabic NLP tasks. For example, generative models like GANs or GPT models can be employed widely for long texts to examine these techniques in such texts.

- **Generalization:** Due to the scarcity of research in this area, we cannot definitively generalize a single method of DA across all contexts. Future research should aim to

develop and test DA methods tailored explicitly for short and long texts, given the unique features of MSA and dialects. When it comes to longer texts, it is essential to ensure that procedures such as synonym replacement, word shuffling, and rule-based approaches are effective without resulting in redundancy or incoherence.

- **Inability to ensure the effectiveness of the DA method:** There is a need for more rigorous evaluations of data generated by DA techniques, specifically for those involving generative models like GPT. Future research should examine the applications of the performance model and the novelties, diversity, and semantics of the generated data rather than just enhancing the performance model. As a result, different data augmentation techniques should be investigated from a more complete perspective.
- **Focus on short texts:** Most current studies on Arabic DA for text classification focus on short texts. Future research should include longer texts, developing and validating DA methods that maintain coherence and relevance over extended passages. Addressing the challenges and advantages of DA techniques will enhance the robustness and applicability of these methods in diverse Arabic NLP tasks.

## CONCLUSIONS

Data augmentation is still emerging in the Arabic context despite extensive exploration of applied DA techniques in other languages' text classification literature. This article comprehensively reviews data augmentation approaches used for the Arabic language in the text classification domain and addresses the five research questions stated early in the introduction. For RQ1, we examined the data augmentation methods employed in Arabic text classification, including paraphrasing, noise addition, rule-based approaches, and data generation. Regarding RQ2, we reported on the data augmentation methods evaluated, particularly those involving data generated by large language models like AraGPT2, and identified the methods used to assess them. Additionally, we discussed the other methods that utilized machine learning or deep learning models to investigate performance after augmentation. For RQ3, we reviewed the domains in which data augmentation techniques have been applied in the context of the Arabic language, noting that sentiment analysis and propaganda detection are the most studied, with four studies each. Furthermore, we highlighted the importance of data augmentation techniques in the Arabic language context, addressing RQ4. Finally, for RQ5, we explored the challenges associated with using data augmentation techniques in the Arabic language.

As a result of a thorough literature analysis, this review underscores the need for further exploration and development of data augmentation approaches for long-term Arabic textual data, leading to potential breakthroughs. Furthermore, there is a need to explain what DA techniques can be used with long or short text or any applicable method for both types. Examining the generated data itself is highly recommended. Applying the augmented data to classifiers to assess their effectiveness is affected by several factors, including the capability of the deep learning or machine learning models, the tuning of their hyperparameters, and the size of the training set, all which contribute to the model's performance. Hence, assessing the quality of the generated data is essential. This review

article provides a foundation for advancing data augmentation techniques in Arabic text classification and aims to inspire further research and innovations in this domain.

## ACKNOWLEDGEMENTS

The authors acknowledge the use of generative AI tools in this work. Specifically, OpenAI's ChatGPT-4 was utilized for proofreading and refining certain parts of the revised manuscript to ensure clarity. The final version was thoroughly reviewed and approved by the authors.

### Funding

Universiti Kebangsaan Malaysia funded this work under the grant TAP-K007009. The funders had no role in study design, data collection and analysis, decision to publish, or preparation of the manuscript.

### Grant Disclosures

The following grant information was disclosed by the authors:
Universiti Kebangsaan Malaysia: TAP-K007009.

### Competing Interests

The authors declare that they have no competing interests.

### Author Contributions

- Samia F. Abdhood conceived and designed the experiments, performed the experiments, analyzed the data, performed the computation work, prepared figures and/or tables, authored or reviewed drafts of the article, and approved the final draft.
- Nazlia Omar conceived and designed the experiments, authored or reviewed drafts of the article, provided the funds, supervision the work, and approved the final draft.
- Sabrina Tiun conceived and designed the experiments, authored or reviewed drafts of the article, supervision the work, and approved the final draft.

### Data Availability

  The article is a literature review.

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
