# Peer review of "Data augmentation for Arabic text classification: a review of current methods, challenges and prospective directions"

_PeerJ Computer Science, doi:10.7717/peerj-cs.2685_

## Round 0.1 · original submission · Major Revisions

All reviewers value the contribution of this literature review to the domain of Arabic NLP, particularly in the use of data augmentation to support text classification. They all however raise a number of concerns that they would like to see addressed in a revision, and hence I recommend the authors perform major revisions following this advice and provide a point-by-point response letter to reviewers with the amendments.

·

Basic reporting

This paper introduces a systematic literature review of data augmentation methods tailored for Arabic text classification. The review is new and falls within the PeerJ journal scope. The authors present an organized introduction to the topic, with detailed discussions of the motivation, target audience, and unique value added of this review in comparison to previous reviews.

Experimental design

The paper presents a well-organized and coherent review of the literature. However, there are some studies that seem relevant based on the set criteria yet are not included in the review. A proper justification is needed to explain why they are not discussed. Here they are:

1. Data Augmentation Techniques on Arabic Data for Named Entity
Recognition (link: https://www.sciencedirect.com/science/article/pii/S1877050921012126)

2. Sarcasm and Sentiment Detection In Arabic Tweets Using BERT-based Models and Data Augmentation (link: https://aclanthology.org/2021.wanlp-1.38/)

3. Arabic dialect identification: An Arabic-BERT model with data augmentation and ensembling strategy (link: https://aclanthology.org/2020.wanlp-1.28/)

4. A Generative Adversarial Network for Data Augmentation: The Case of Arabic Regional Dialects (link: https://www.sciencedirect.com/science/article/pii/S1877050921011674)


5. Enhancing Detection of Arabic Social Spam Using Data Augmentation and Machine Learning (link: https://www.mdpi.com/2076-3417/12/22/11388)


6. Multi-Task Learning Model with Data Augmentation for Arabic Aspect-Based
Sentiment Analysis (link: https://cdn.techscience.cn/files/cmc/2023/TSP_CMC-75-2/TSP_CMC_37112/TSP_CMC_37112.pdf )

Validity of the findings

The authors systematically developed the content to answer the research questions set in the introduction, making it easy to get the answers, as well as current gaps and future directions.

Additional comments

Comments on text:
1. The citing style in the paper needs proper modifications. You need to cite the authors or papers based on the context. For example, in line 298, “(Bensoltane & Zaki, 2023)” should be citing the authors, i.e., Bensoltane & Zaki (2023). This needs to be updated in the whole paper.
2. In line 303, you are referring to examples in Table 2 to show examples of the paraphrasing technique used in the paper (Bensoltane & Zaki, 2023). However, the examples are brought from a different paper. Can you explain why?
3. The position of the title on line 390 at the end of the page looks weird and needs to be re-positioned properly. Also, replacing the word ‘evaluation’ with ‘evaluating’ seems more relevant to the content.
4. Are the improvements discussed between lines 453-474 pertain to the same task/dataset? You mentioned the improvements in F1 score but didn’t discuss whether these improvements belong to the same task or dataset.
5. In line 256, you mentioned: “Figure 1 shows the growing interest and escalating research activity in the field over the past six years”. However, the figure does not match this statement, as the number of studies went down between 2023 and 2024. Besides, I feel it is a bit strange to claim escalating research activity based on 13 studies only.
6. Why do the captions of all figures and tables start with Table 1 or Figure 1? You need to correct this.

Comments on Figures:
1. The resolution of all figures is quite low. So, it is necessary to improve the resolution in order to make the figures more readable.

Comments on Tables:
a. In Table 1, you used TC without defining it. Do you mean ATC?
b. Table 3 can be made more compact by removing extra unused space and eliminating the last column as it can be easily captured from the previous column. Same applies to table 4.
c. You can save wide space in Table 6 by explaining the first column in the caption.
d. Table 8 is quite dense, with some wasted space in the first column. Instead, you make the text vertical in the first column and allow more space for other columns. Also, try to balance the width of the columns so the empty space is minimized as possible. Note the big empty space in the first row of the “Noise”.
e. In the “Insertion some words” row, you mentioned “performance between 15-21 non-augmented data”. What measure are you referring to here?
f. In the improvement column of Table 8, it is not clear whether the improvements mentioned are on the same dataset or task. Please elaborate here to make a fair comparison.
g. Table 8 contains some repetition. For example, there are two rows for “Generate new text/document” with nearly the same content. The only difference is the used model and improvements, which can be merged together.
h. What does “[63]” mean after “Achieved 93.8% accuracy, 93.3% macro F1, and 93.84% weighted F1” in Table 8?
i. What does “[9]” mean in the last line of Table 8?

Reviewer 2 ·

Basic reporting

The paper provides a survey and overview of data augmentation methods in Arabic text classification. Generally, the paper is clear and easy to read and follow. Such papers would be really useful in the Arabic NLP community, and I want to thank the authors for their efforts. The structure of the paper is sound, but I am not sure why the authors did not include the figures and tables within the actual manuscript part.

I think the structure of the paper needs improvement. Basing it on the provided research questions is slightly confusing. It would have been better to revisit the questions towards the end of the paper and provide their answers.

Experimental design

Regarding the methodology, the authors use well-known databases to identify relevant papers. The authors identify 13 papers out of 281. Although the authors consider 4 databases, I don't think this provides good coverage of the Arabic NLP literature. The authors should have included other sources that would include relevant Arabic NLP papers. The main one would be ACL Anthology, which is the main repository for top-tier NLP conferences. Also, given ACL Anthology is not included, the papers from ArabicNLP (Arabic NLP conference, formerly WANLP) and OSACT are missing. These two venues have been very important for the community in the last decade, and failure to include them is a major weakness of the paper.

Additionally, it is not clear why authors limited the scope of the methodology to papers related to text classification. How would data augmentation in other tasks be different? Nevertheless, given the small number of relevant papers. I think expanding the scope of the study would have been really useful and enriching for the paper.

Validity of the findings

The discussion in the paper is slightly shallow. The paper would benefit from a deeper dive into the discussed approaches and a comparison of them, identifying possible advantages/disadvantages for each of them.

Reviewer 3 ·

Basic reporting

The paper is written in generally clear and understandable English. However, we note some titles which seem poorly formulated :
Line 91: Studying Motivation
Line 390: Evaluation Data Generated by Generation Models (Evaluation of ...)
Line 402: Evaluation Data Created by Transformation Methods (Evaluation of ...)

Tables: All given tables have a unique number (Table 1).
Figures: All given figures have a unique number (Figure 1).

Experimental design

Line 394: ÷ Similarity Scores : No need to define a bullet point if that is the only point to be presented. Doing so suggests that other methods of evaluation will follow.

In the "DISCUSSION" section (line 424), we believe that the "Data Augmentation Method" subsection (from line 436 to line 451) is a redundancy with the "RQ1" subsection of the "ANALYSIS" section (line 247). In this part of the discussion, it would have been more interesting to propose a synthesis of the advantages and disadvantages of the DA methods as well as an overview of their performance, that is to say to propose a discussion/interpretation of table 8.

Validity of the findings

In their interpretation of Table 5 (about the evolution of DA methods in Arabic text classification (ATC), the authors specify in lines 376 and 377 that “DA in ATC emerged in 2019 with noise methods…”. How can the authors affirm this if their study begins in 2019? Did they study articles dating from before 2019 in order to verify that the first DA works in ATC date only from 2019 ?

In table 6 which summarizes the part of the study concerning DA evaluation, The meaning and content of the column named “methods” in the table are not clear ? is it the type of evaluation (evaluation of augmented data vs evaluation of the classification model using augmented data? In this case, it should be clearly shown in the table.

In the subsection “RQ3: Domain” (line 484), the authors provide different cases of NLP applications treated in the literature and in which data augmentation was used (Sentiment Analysis, question prediction, dialect identification, etc.). The discussion provided by the authors on this point (see paragraph between lines 507 and 516), suggests that specific data augmentation methods would be more appropriate than others for each type of application (they cite for example, GAN for Dialect identification, using rules for question prediction and detection). This should be considered with much more caution by the authors. I think indeed, that the very small number of papers examined, dealing with each of these applications, does not allow to have a clear vision on the best techniques that can be appropriate to each type of application. It is necessary to examine much more DA works in these application areas in order to conclude on the best techniques to adopt for each.

The study of the datasets used in the reviewed articles, should provide additional information, not provided in the present study and which relates to the availability or not of the dataset. This information is very useful for researchers wishing to work on these topics in Arabic NLP and who would need datasets as benchmarks.

---

## Round 0.2 · Minor Revisions

One of the original reviewers has reassessed the submission, and based on my own read as well, we agree that it has improved substantially and that it is progressing towards acceptance.

Before it can be accepted, the minor revisions raised by the reviewer need to be addressed.

·

Basic reporting

This paper introduces a systematic literature review of data augmentation methods tailored for Arabic text classification. The review is new and falls within the PeerJ journal scope. The authors present an organized introduction to the topic, with detailed discussions of the motivation, target audience, and unique value added to this review in comparison to previous reviews.

Experimental design

After addressing the reviewers' comments, the authors added multiple relevant studies to cover the literature in the suggested time span, which added valuable insights to the paper.

Validity of the findings

No comment

Additional comments

I noted some differences in the .pdf and .docx files. Therefore, my comments are based on the pdf version:
- Convert the citation style in line 50 to paper citation, not author citation.
- Convert the citation style to paper style in line 433.
- It is not clear in Figure 2 how many papers are included in the survey. It would be better to add a point on the line aligned with each year to illustrate the number of articles included in that year.
- In lines 468-470, you need to mention what measures (accuracy, precision, etc.) were used to measure the improvements of DA, not just the model's names like BERT-based models and some classifiers.

---

## Round 0.3 · accepted · Accept

The paper can now be accepted following the last round of minor revisions.